# Systemic inflammation is associated with worse outcomes from SARS-CoV-2 infection but not neutralizing antibody

Christopher W. Farnsworth,[1] Brittany Roemmich,[1] John Prostko,[2] Gerard Davis,[2] Gillian Murtagh,[2] Laurel Jackson,[2] Christopher Jacobson,[2] Nicolette Jeanblanc,[2] Timothy Griffiths,[2] Edwin Frias,[2] David J. Daghfal[2]

**ABSTRACT** Systemic inflammation is associated with COVID-19 mortality rates, but the impact of inflammation on neutralizing antibodies to severe acute respiratory syndrome-related coronavirus 2 (SARS-CoV-2) and on outcomes is poorly understood. This study aimed to determine the association between neutralizing antibody responses, inflammation, and clinical outcomes in hospitalized patients with COVID-19. Two hundred and eight patients presenting to the ED with symptomatic SARS-CoV-2 were included. Neutralization was assessed using the architect angiotensin-converting enzyme-2 (ACE2) binding inhibition assay, and inflammation was assessed using C reactive protein (CRP) and interleukin 6 (IL-6). Medical records were examined for 30-day mortality and 10-day intubation. Correlation between biomarkers was assessed and Kaplan–Meier curves and Cox proportional hazards models were constructed for outcomes. Thirty-seven (18%) patients died and 59 (28%) required intubation. There was a correlation between IL-6 and CRP ($r = 0.34$) but not ACE-2 ($r < 0.06$). Patients that died had higher CRP (14 mg/dl, 8–21) than those that survived (5 mg/dl, 2–11) and IL-6 (died = 344 pg/ml, 138–870 vs. survived = 65 pg/ml, 28–140). ACE-2 inhibition trended higher in those who survived (18%, 0%–65%) than those who died (3%, 0%–48%). Patients with elevated IL-6, elevated CRP, or low ACE2 inhibition had higher mortality. Only IL-6 (hazard ratio: 1.28, 95% CI 1.08–1.52) and age (1.04, 1.01–1.08) were associated with mortality in multivariate models. Elevated IL-6 was associated with 30-day mortality from SARS-CoV-2 infection. Lower ACE-2 inhibition was not independently associated with mortality or correlated with inflammatory markers, implying the importance of other aspects of the immune response for reducing SARS-CoV-2 mortality risk.

**IMPORTANCE** While systemic inflammation associated with worse outcomes from SARS-CoV-2 infection, it is not associated with neutralizing antibody concentrations, implying the importance of other aspects of the immune response for reducing SARS-CoV-2 mortality risk.

**KEYWORDS** inflammation, adaptive immunity, neutralizing antibodies, SARS-CoV-2

Numerous studies have demonstrated an association between excessive systemic inflammation and mortality in patients with COVID-19. To this end, blood concentrations of both C-reactive protein (CRP) and interleukin-6 (IL-6) have been associated with worse outcomes including mortality (1–3). Studies have demonstrated that genetic variants in the IL-6 inflammatory pathway are associated with mortality from COVID-19 (4). Furthermore, IL-6 concentrations correlate with other markers of systemic inflammation including CRP and soluble tumor necrosis factor receptor 1 (TNFR1) in patients with acute respiratory distress syndrome (ARDS) from COVID-19 (5). Inhibition of inflammation using therapies such as corticosteroids and IL-6 antagonists has been shown to

**Peer Reviewer** Jonathan Daniel Hulse, Shepherd University, Shepherdstown, West Virginia, USA

Address correspondence to Christopher W. Farnsworth, cwfarnsworth@wustl.edu.

C.W.F. received research funding from Abbott for portions of this study. L.J., J.P., G.D., G.M., C.J., N.J., T.G., E.F., and D.J.D. are all employees of Abbott Diagnostics.

See the funding table on p. 9.

reduce mortality in some studies (6, 7). Together, this implies an important role of a hyper-inflammatory host immune response in worse outcomes from COVID-19.

While systemic inflammation is associated with worse outcomes in patients with COVID-19, it is unclear how neutralizing antibodies to severe acute respiratory syndrome-related coronavirus 2 (SARS-CoV-2), particularly in naïve hosts, associates with outcomes. Numerous studies have demonstrated an association between neutralizing antibody titers and protection after vaccination or infection (8–11). However, in patients hospitalized for COVID-19 that have not previously been exposed to the SARS-CoV-2 virus or vaccine, the importance of the neutralizing antibody response is poorly defined. Previous studies have demonstrated that increased neutralization potency to SARS-CoV-2 occurs in patients with more severe COVID-19-related symptoms and that higher neutralizing antibody titers are associated with better outcomes (12). Consistent with this, patients receiving neutralizing monoclonal antibodies to SARS-CoV-2 have a reduced risk of hospitalization and death in both unvaccinated patients and patients who are immunocompromised (13). Further, how systemic inflammation impacts neutralizing antibody production following SARS-CoV-2 infection is relatively poorly understood. One previous study demonstrated that patients with severe COVID-19 infection had higher IL-6 concentrations and antibody titers relative to healthy controls and those with mild COVID-19 (14). However, several patients that died with severe COVID-19 had relatively low antibody titers and very high IL-6 concentrations, implying an inverse correlation between the inflammatory response and neutralizing antibody titer.

The primary aim of this study was to determine the association between the neutralizing antibody response to SARS-CoV-2, inflammatory markers, and clinical outcomes in a cohort of hospitalized patients without prior exposure/vaccination. The secondary objective was to assess how the serial change in neutralizing antibodies associates with outcomes in patients hospitalized with COVID-19.

## MATERIALS AND METHODS

### Study design and patient samples

This prospective observational study approved by the Washington University IRB (#202005042 and #202303108) under a waiver of consent included 208 adults admitted through the Emergency Department (ED) at Barnes Jewish Hospital between 1 October 2020 and 29 October 2021 with symptomatic RT-PCR confirmed SARS-CoV-2 infection without previous vaccination or documented exposure to SARS-CoV-2. Remnant EDTA specimens were procured within 24 h of collection. In a subset of patients, serial samples were obtained, totaling 395 specimens. Specimens were stored at 4°C for the first 24 h before being frozen in aliquots at −80°C for up to two years before testing. The electronic medical record (EPIC) was examined between 10 January 2020 and 01 January 2021 for comorbidities, age, gender, 30-day mortality, requirement for 10-day intubation, and pharmacological interventions by two observers blinded to biomarker results. Patient identifiers were retained and removed at the earliest opportunity.

### Testing

Specimens were thawed and tested for angiotensin-converting enzyme-2 (ACE2) inhibition, CRP, and IL-6. Neutralization of SARS-CoV-2 was assessed using the architect ACE2 binding inhibition assay (Abbott, Research Use Only) which measures the percent inhibition of interactions between antibodies and the SARS-COV-2 viral spike receptor binding domain. The chemiluminescent assay incubates serum or plasma samples with paramagnetic particles coated with the spike RBD protein. The initial incubation is followed by a wash and then the addition of ACE2-recombinant antigen labeled with acridinium. The signal is then generated utilizing peroxide and base solutions that trigger the remaining acridinium moiety in the reaction. The relative light units (RLUs)

are measured and read off a calibration curve for quantitation. An inhibition of 12.5% of ACE2 correlates to >80% neutralization of SARS-COV-2 by plaque reduction neutralization assays (15–17). CRP was assessed using the architect multigent CRP assay with 9.6 mg/dl as the categorical cutoff for risk (18, 19). The ARCHITECT MULTIGENT CRP Vario is a latex immunoassay that measures CRP in serum or plasma. When a reaction occurs between the CRP in a sample and the anti-CRP antibody coated on latex particles, agglutination results. The agglutination is detected as a change in absorbance at 572 nm. The rate of absorbance change is proportional to the quantity of CRP in the sample. The CRP assay used has a normal range of ≤0.5 mg/dl and a measuring interval of 0.02–48.00 mg/dl. The imprecision of the CRP assay is 0.7% at a concentration of 8.2 md/dl. IL-6 was assessed using the Alinity IL-6 assay (Abbott, Research Use Only) with 30 pg/ml as the threshold for risk in categorical analyses (20). The Alinity IL-6 assay is a two-step sandwich immunoassay. In the first step, the specimen is incubated with anti-IL-6 antibody-coated paramagnetic microparticles. IL-6 present in the specimen binds to the anti-IL6 antibody-coated microparticles. After washing, anti-α IL-6 acridinium labeled conjugate is added in the second step. Pre-trigger and trigger solutions are then added to the reaction mixture; the resulting chemiluminescent reaction is measured as RLUs. A direct relationship exists between the amount of IL-6 in the reaction and the RLUs detected by the Alinity I optical system and is quantified from a calibration curve. The imprecision of the IL-6 assay was 5.2% at 9.8 pg/ml and 4.5% at 3,480 pg/ml.

## Statistics

Descriptive statistics were summarized for the patient population overall. For continuous variables, mean and standard deviation were reported if approximately normal distribution and median and interquartile range (IQR) otherwise. Categorical variables were summarized by number and percentage. Correlation between biomarkers was assessed using Pearson's $r$. Kaplan–Meier curves and Cox proportional hazard models were constructed for 30-day mortality and 10-day intubation. Pairwise comparisons were performed using the Mann–Whitney U test. Log-rank tests were performed to determine the significance between groups in survival analyses. Differences in ACE2 inhibition between those who survived and died from serial testing were assessed using the Wilcoxon rank sum test. All statistics were performed using R version 4.3.0 (21).

**TABLE 1** Patient demographics

| Characteristic | $N = 208^a$ |
|---|---|
| Symptom onset to baseline (days) | 3 (2, 7) |
| Unknown | 1 |
| Age | 66 (55, 72) |
| Sex | |
| Female | 84 (40%) |
| Male | 124 (60%) |
| Race | |
| Asian | 4 (2.0%) |
| Black | 150 (75%) |
| White | 45 (23%) |
| Unknown | 9 |
| BMI category | |
| Normal | 64 (32%) |
| Obese | 84 (41%) |
| Overweight | 42 (21%) |
| Underweight | 13 (6.4%) |
| Unknown | 5 |
| 30-day mortality | 37 (18%) |
| 10-day intubation required | 59 (28%) |

$^a$Median (IQR); $n$ (%).

## RESULTS

In total, 208 patients were enrolled that had a baseline measurement available within 5 days of admission. The median time to symptom onset prior to presentation to the ED was 3 days (IQR:2–7) and the median age was 66 (55–72, Table 1). Seventy-five percent of the cohort self-identified as black, 23% identified as white, and 2% identified as Asian, with nine patients having unknown race. Thirty-seven of 208 (18%) patients died within 30 days of ED presentation and 59 (28%) required intubation within 10 days.

There was a weak correlation between IL-6 and CRP (Pearson $r = 0.34$, $P$-value < 0.001) but not with ACE2 inhibition (Pearson $r < 0.06$ for both IL-6 and CRP, Fig. 1). There was significantly higher CRP in those that died (median = 14 mg/dl, IQR = 8–21) than those that survived (5 mg/dl, 2–11, Fig. 2A). IL-6 was higher in patients that died (344 pg/ml, 0–870) than those that survived (65 pg/ml, 28–140, Fig. 2B). Conversely, the median ACE2 inhibition results trended higher in patients that survived (18%, 0%–65%) versus patients that died (3%, 0%–48%, Fig. 2C). Similar trends were observed for 10-day intubation (Fig. S1).

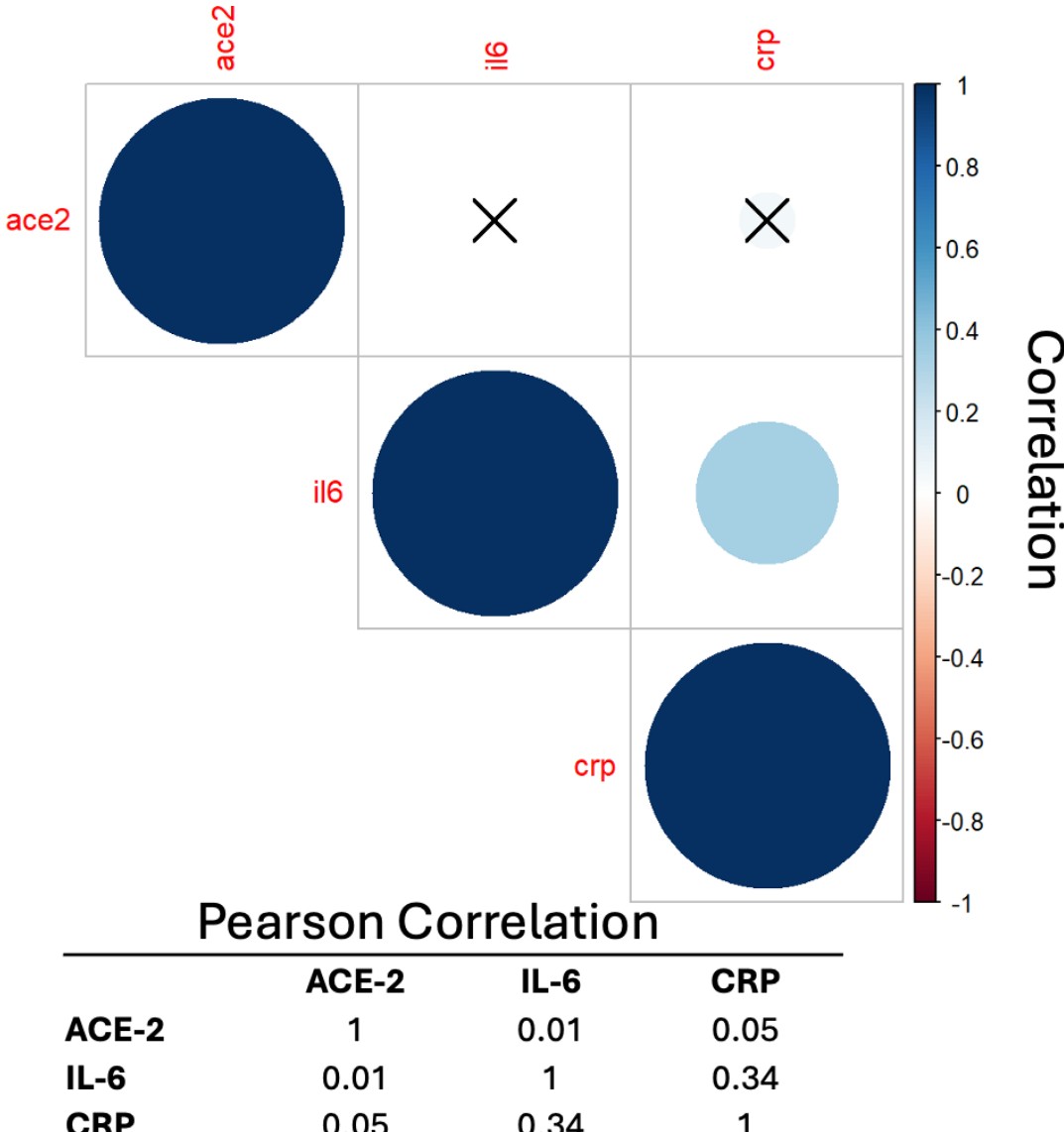

| Pearson Correlation | | | |
|---|---|---|---|
| | **ACE-2** | **IL-6** | **CRP** |
| **ACE-2** | 1 | 0.01 | 0.05 |
| **IL-6** | 0.01 | 1 | 0.34 |
| **CRP** | 0.05 | 0.34 | 1 |

**FIG 1** Correlation between IL-6, CRP, and ACE-2 inhibition in patients hospitalized with COVID-19. Shown in the box are Pearson's $r$ for each pair of analytes. Circles with a black x have no correlation.

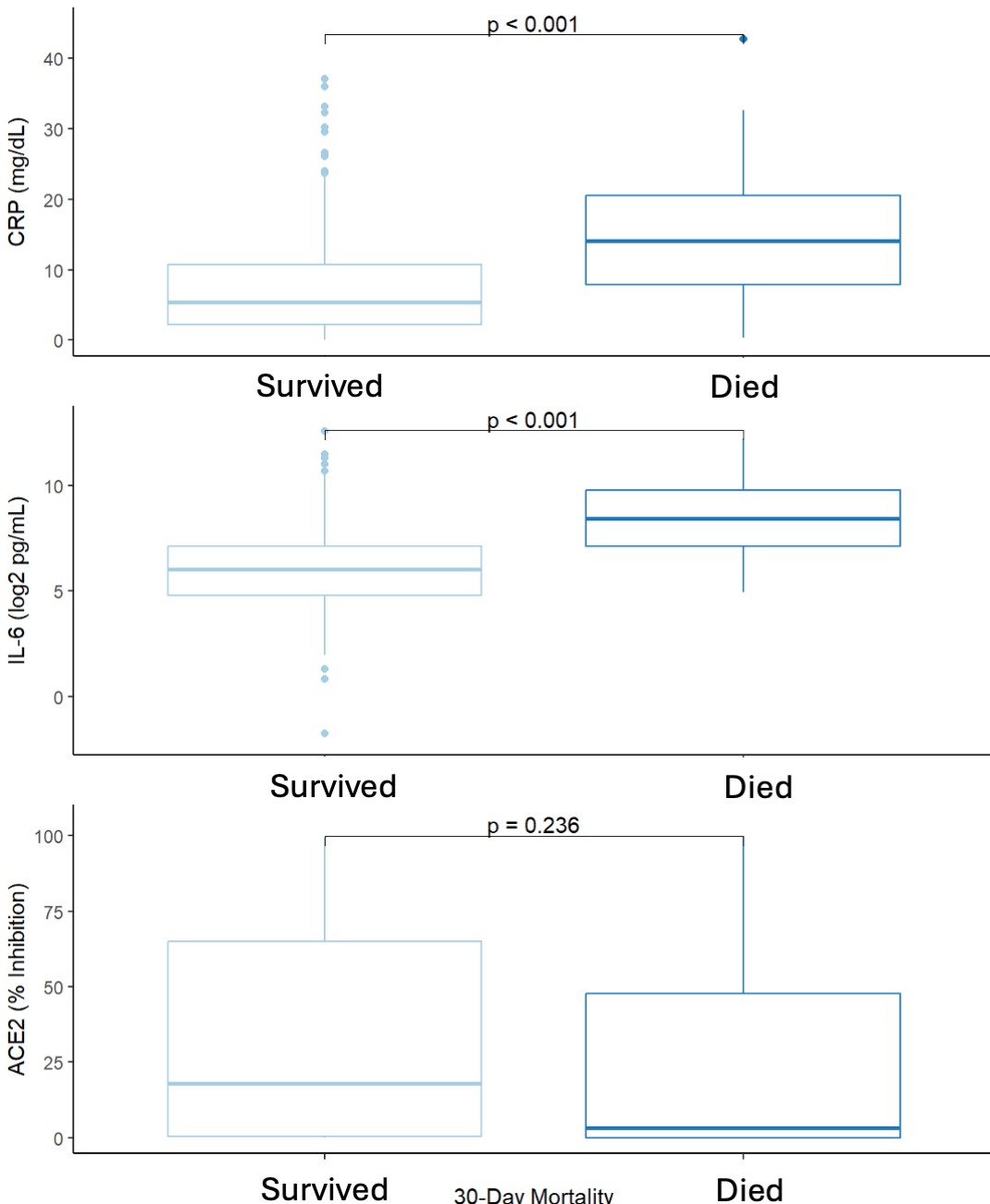

**FIG 2** Concentrations of CRP, IL-6, and ACE-2 inhibition in patients hospitalized with COVID-19 in those that survived (red) vs. those that died (blue). Lines are at the median, boxes represent the IQR, and whiskers are the central 95%.

Survival curves demonstrated that patients with IL-6 ≥30 pg/ml (*P* = 0.0096), CRP ≥9.6 mg/dl (*P* = 0.023), and ACE2 binding <12.5% (*P* = 0.031) at baseline were more likely to die within 30 days (Fig. 3). Similar trends were observed for 10-day intubation, with IL-6 ≥30 (*P* = 0.006) and CRP ≥9.6 (*P* = 0.0051) significantly stratifying risk of intubation but not ACE2 (Fig. S2).

Multivariate Cox regression modeling found that IL-6 concentrations were significantly associated with a 30-day mortality with a hazard ratio (HR) of 1.28 (95%CI 1.08–1.52) for each doubling (Table 2). Increased age was also significantly associated with 30-day mortality with a HR of 1.04 (1.01–1.08). ACE2, CRP, sex, and symptom onset time were not significantly associated with 30-day mortality. Only IL-6 was significantly

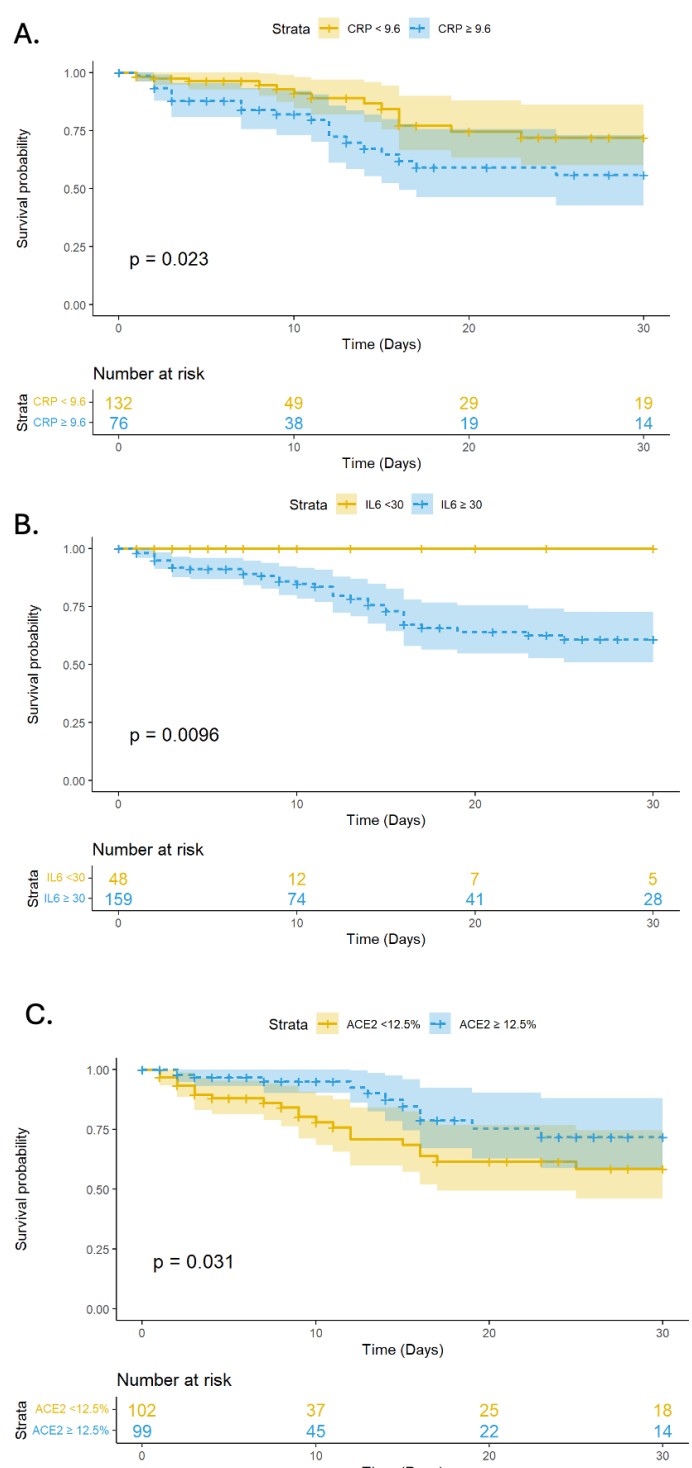

**FIG 3** Kaplan–Meier survival curves for 30-day mortality in those below (yellow) and above (blue) designated thresholds for (A) CRP, (B) IL-6, and (C) ACE-2 inhibition.

associated with an increased risk of requiring intubation within 10-day, with a HR of 1.47 (1.29–1.68, Table S1) per doubling in concentration. Sensitivity analyses were also performed for both outcomes adjusting for race (excluding the nine patients with unknown race) and observed significant associations held (Table S2). Serial changes in ACE2 inhibition were assessed in 85 patients who survived and 20 patients who died (Fig. 4). Patients who survived had a median increase in ACE2 inhibition of 1.63% per day

**TABLE 2** Cox proportional hazard models for 30-day mortality

| Characteristic | N | HR[a] | 95% CI[b] | P-value |
|---|---|---|---|---|
| ACE-2 | 200 | 0.99 | 0.98, 1.00 | 0.2 |
| log2(IL-6) | 200 | 1.28 | 1.08, 1.52 | 0.004 |
| log2(CRP) | 200 | 1.12 | 0.85, 1.48 | 0.4 |
| Age | 200 | 1.04 | 1.01, 1.08 | 0.006 |
| Sex | 200 | | | |
| Female | | — | — | |
| Male | | 0.69 | 0.34, 1.40 | 0.3 |
| Symptom onset (Days) | 200 | 0.97 | 0.92, 1.02 | 0.2 |

[a]HR = Hazard Ratio.
[b]CI = Confidence Interval.

(IQR: 0.22–4.7), and those who died had an increase in ACE2 inhibition of 2.32% per day (0.0–7.79, P = 0.5).

## DISCUSSION

Patients with systemic inflammation associated with COVID-19 infection are at known risk for worse outcomes (1, 3, 4, 14). In this study, we hypothesized that systemic inflammation (as measured by IL-6 and CRP) impacts the production of neutralizing antibodies to SAR-CoV-2 and ultimately influences the probability of survival. We found that while low neutralizing antibodies to the ACE-2 receptor at baseline were associated with worse outcomes, ACE-2 inhibition was not an independent predictor of mortality, and it was not associated with IL-6 and CRP. In contrast, IL-6 concentrations (but not CRP) were independently associated with mortality and requirement of intubation and strongly predicted survival. Together, these results imply that there may be a limited association between antibody production, inflammation, and survival in COVID-19.

Consistent with previous studies, we found that both CRP and IL-6 were associated with worse outcomes from COVID-19 including mortality and requirement for 10-day intubation. Of note, despite both IL-6 and CRP being higher in patients with worse outcomes, the correlation between the biomarkers was only 0.34. This is similar to a previous study in patients with ARDS from COVID-19, which demonstrated a correlation of 0.46 between IL-6 and CRP (5). This may imply that each biomarker provides different information regarding the biology or pathophysiology of the host immune response. To this end, only IL-6 was significantly associated with mortality and requirement for

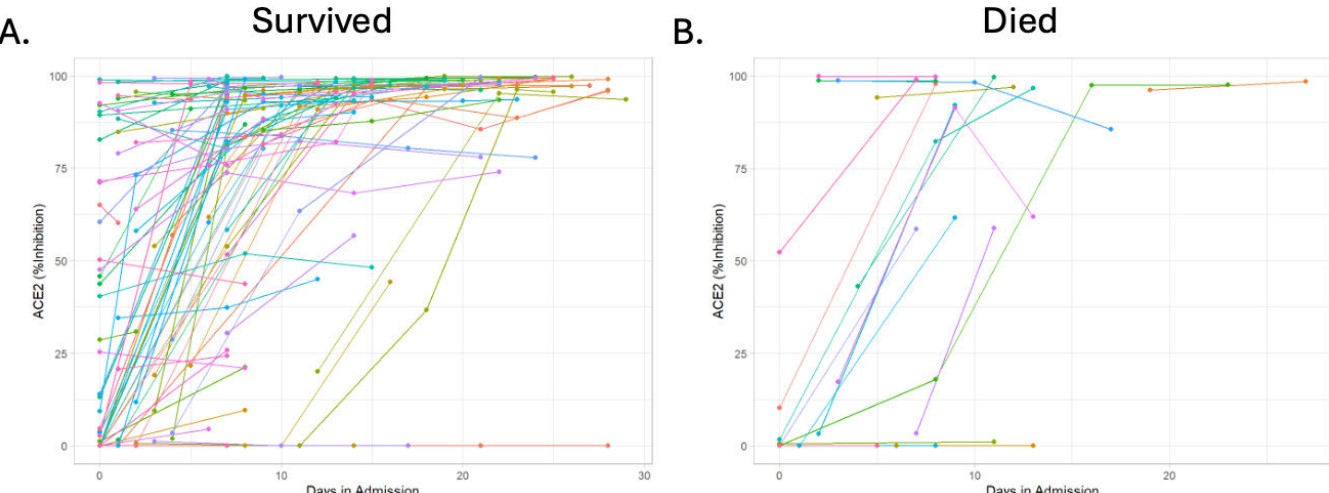

**FIG 4** Serial neutralization testing was performed in patients that (A) survived or (B) died within 30 days of admission. Each line connecting two or more points represents an individual patient.

10-day intubation in the multivariate models. This study further supports the use of IL-6 to discriminate patients with worse outcomes from COVID-19 and may be useful for identifying those at the highest risk for worse outcomes.

The literature is mixed with regard to the importance of neutralizing antibody production to SARS-CoV-2 in naïve hosts and outcomes. While it is clear from studies that prophylaxis with monoclonal antibodies and vaccination protect the host from worse outcomes such as mortality (8, 13, 22), studies in naïve hosts have been less clear. A previous study of 113 SARS-CoV-2 infected patients found that patients who died or required intubation had higher CRP, IL-6, and higher neutralizing antibody titers than non-hospitalized patients with COVID-19 (12). However, the total number of hospitalized patients was relatively small ($n$ = 82), the time of specimen collection was unclear, and no difference in neutralizing antibody titer was observed in those hospitalized with worse outcomes relative to the remainder of the hospitalized cohort. A strength of our study was that the cohort was relatively large ($n$ = 208), all presenting to the ED with a specimen drawn within the first 5 days of admission. Thus, the differences observed in this study with regard to the lack of association between IL-6 and neutralizing antibodies may be due to the population assessed. Nonetheless, this study implies that neutralizing antibody concentrations as measured by ACE2 inhibition may not be appropriate for adjudicating the use of monoclonal antibody therapies in patients critically ill with COVID-19.

There were multiple limitations of this study. While time from symptom onset to admission was noted for each patient, the variable time to presentation may have dramatically impacted the serological response. Furthermore, due to volume limitations, we were not able to assess anti-SARS-CoV-2 IgG. Previous studies have shown that the ratio of neutralizing antibodies to anti-Spike IgG better correlates with protection and outcomes than neutralizing antibodies alone (12). While the study size here is one of the largest presenting to the ED in the literature to our knowledge, it was still limited to only 38 (18%) mortalities and 59 (28%) patients requiring intubation. Another limitation is that this study occurred in the early waves of the pandemic during the rise of the B.1.1.7 alpha strain and the B.1.351 strain. During this time, there was limited prior immune protection, high virulence, and high mortality. As a result, this study may overestimate the HRs in vaccinated individuals with contemporary SARS-CoV-2 strains.

## Conclusion

Systemic inflammation measured by IL-6 upon admission is highly associated with 30-day mortality from SARS-CoV-2 infection. While ACE-2 inhibition was lower in those who died from COVID-19 at admission, it was not independently associated with mortality or correlated with inflammatory markers. This implies the importance of other aspects of the immune response for mediating inflammation and reducing the risk of mortality from SARS-CoV-2.

## AUTHOR AFFILIATIONS

[1]Department of Pathology and Immunology, Washington University in St. Louis School of Medicine, St. Louis, Missouri, USA
[2]Abbott Diagnostics, Abbott Park, Illinois, USA

## AUTHOR ORCIDs

Christopher W. Farnsworth ⓘ http://orcid.org/0000-0001-7169-3850

## FUNDING

| Funder | Grant(s) | Author(s) |
|---|---|---|
| Abbott \| Abbott Diagnostics (DIAGNOSTICS AT ABBOTT) | | Christopher W. Farnsworth |

## AUTHOR CONTRIBUTIONS

Christopher W. Farnsworth, Conceptualization, Data curation, Methodology, Writing – original draft | Brittany Roemmich, Data curation | John Prostko, Formal analysis, Investigation, Writing – original draft | Gerard Davis, Formal analysis | Gillian Murtagh, Conceptualization, Writing – original draft | Laurel Jackson, Formal analysis | Nicolette Jeanblanc, Conceptualization, Writing – original draft | Timothy Griffiths, Conceptualization, Writing – original draft | Edwin Frias, Conceptualization, Writing – original draft.

## DATA AVAILABILITY

Data will be made available on request to the corresponding author.

## ADDITIONAL FILES

The following material is available online.

### Supplemental Material

**Supplemental material (Spectrum02459-24-s0001.pdf).** Fig. S1 and S2; Tables S1 and S2.

### Open Peer Review

**PEER REVIEW HISTORY (review-history.pdf).** An accounting of the reviewer comments and feedback.

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
