## [Reviewer comments · Microbiology Spectrum]

Microbiology Spectrum

Systemic Inflammation is associated with worse outcomes from SARS-CoV-2 infection but not neutralizing antibody

Christopher Farnsworth, Brittany Roemmich, John Prostko, Gerard Davis, Gillian Murtaugh, Laurel Jackson, Christopher Jacobson, Nicolette Jeanblanc, Timothy Griffiths, Edwin Frias, and David Daghfal

Corresponding Author(s): Christopher Farnsworth, Washington University in St Louis Washington University Physicians

Review Timeline:

Submission Date:	September 30, 2024
Editorial Decision:	November 12, 2024
Revision Received:	December 11, 2024
Accepted:	December 16, 2024

Editor: Jie Wang

Reviewer(s): Disclosure of reviewer identity is with reference to reviewer comments included in decision letter(s). The following individuals involved in review of your submission have agreed to reveal their identity: Jonathan Daniel Hulse (Reviewer #1)

Transaction Report:

DOI: <https://doi.org/10.1128/spectrum.02459-24>

Re: Spectrum02459-24 (Systemic Inflammation is associated with worse outcomes from SARS-CoV-2 infection but not neutralizing antibody)

Dear Dr. Christopher W. Farnsworth:

Thank you for the privilege of reviewing your work. Below you will find my comments, instructions from the Spectrum editorial office, and the reviewer comments.

Revision Guidelines

Sincerely,
Jie Wang
Editor
Microbiology Spectrum

Reviewer #1 (Comments for the Author):

This manuscript adds to the growing body of work associated with COVID-19/SARS-CoV-2. This research adds to the growing knowledge surrounding inflammatory response associated with SARS-CoV-2 infection. The overall writing is clear and concise, with excellent english and view few errors.

There are a few suggestions below:

Line 34: Is interrogated the best word choice?

Line 100: Missing the degree symbol before C

Line 101: Is interrogated the best word choice?

Line 178: Why is the data not shown? What is the association of race and the statistical outcomes?

Other Questions:

Was the data broken down further (table 1) to show percents of each race in respect to BMI?

The figures have inconsistent labeling.

Figure 1 missing labels

Figure 2 says 'yes' and 'no'. What does this mean? Missing labels.

Figure 3 should probably label each chart as a different portion of figure 3. Ex: 3a, 3b, 3c, etc.

Reviewer #2 (Comments for the Author):

Farnsworth et al. evaluated laboratory markers (IL-6, CRP), humoral response (ACE2-binding inhibition), and the COVID-19 outcome when presenting to the ED and administrating.

Overall results presented are mostly well-known (PMID: 32294485, 32839624, 32845042, 33201896, 33412089, 33753738, etc.) and the potential impact of the manuscript is limited. On the other hand, the current cohort includes much higher mortality and intubation rates during periods of Alpha and Delta variant emergence (PMID: 35085223). This shall reflect a potential selection bias, which can over- or under-score the hazard ratio analyzed. Further, as the authors discussed, the sampling time primarily affected the result when only a single data point was analyzed. Therefore, while the result presented in the current study follows the previous research, careful attention is needed to discuss the result.

Rather than following the well-known predictors of severe COVID-19 outcomes, I suggest focusing on evaluating such severe patients to turn the selection bias into the study's originality with detailed medical record analysis (at least well-known predictors and treatment) and, ideally, additional serial change in markers and humoral response.

Minor Comments:

1. Figure 1.

I prefer to present scatterplots to show the correlation between the factors, including "x," which displays columns. Also, Pearson's r value between 0.2 and 0.4 is generally considered a weak or poor correlation. Do not describe "significant correlation" only due to the p-value in correlation analysis, which is mostly misleading.

2. The statistics method of Fig. 2/Sup. Fig. 2 needs to be included.

Thank you to both reviewers whose recommendations have added significant clarity to this manuscript.

Reviewer #1 (Comments for the Author):

This manuscript adds to the growing body of work associated with COVID-19/SARS-CoV-2. This research adds to the growing knowledge surrounding inflammatory response associated with SARS-CoV-2 infection. The overall writing is clear and concise, with excellent english and view few errors.

There are a few suggestions below:

Line 34: Is interrogated the best word choice?

Thank you, this has been modified to examined at both instances

Line 100: Missing the degree symbol before C

Thank you, this has been added for both 4°C and -80 °C

Line 101: Is interrogated the best word choice?

Thank you, this has been modified to examined

Line 178: Why is the data not shown? What is the association of race and the statistical outcomes?

Thank you for the opportunity to clarify. This has been added as supplemental table 2. There was no different in statistical outcomes.

Other Questions:

Was the data broken down further (table 1) to show percents of each race in respect to BMI?

Thank you for this suggestion. It is an interesting consideration but given the relatively low number of patients, breaking down by both race and BMI resulted in an underpowered analysis that was difficult to interpret.

The figures have inconsistent labeling.

Figure 1 missing labels

Figure 2 says 'yes' and 'no'. What does this mean? Missing labels.

Figure 3 should probably label each chart as a different portion of figure 3. Ex: 3a, 3b, 3c, etc.

Thank you, labeling has been updated for all as suggested by the reviewer.

Reviewer #2 (Comments for the Author):

Farnsworth et al. evaluated laboratory markers (IL-6, CRP), humoral response (ACE2-binding inhibition), and the COVID-19 outcome when presenting to the ED and administrating.

Overall results presented are mostly well-known (PMID: 32294485, 32839624, 32845042, 33201896, 33412089, 33753738, etc.) and the potential impact of the manuscript is limited. On the other hand, the current cohort includes much higher mortality and intubation rates during periods of Alpha and Delta variant emergence (PMID: 35085223). This shall reflect a potential selection bias, which can over- or under-score the hazard ratio analyzed. Further, as the authors discussed, the sampling time primarily affected the result when only a single data point was analyzed. Therefore, while the result presented in the current study follows the previous research, careful attention is needed to discuss the result.

We thank the reviewer for this feedback and agree with the interpretation. We have added the following as a limitation

“Another limitation is that this study occurred in the early waves of the pandemic, during the rise of the B.1.1.7 Alpha strain and the B.1.351 strain. During this time, there was limited prior immune protection, high virulence, and high mortality. As a result, this study may overestimate the hazard ratios in vaccinated individuals with contemporary SARS-CoV-2 strains.”

Rather than following the well-known predictors of severe COVID-19 outcomes, I suggest focusing on evaluating such severe patients to turn the selection bias into the study's originality with detailed medical record analysis (at least well-known predictors and treatment) and, ideally, additional serial change in markers and humoral response.

We thank the reviewer for this suggestion and agree that the data here is not particularly novel. The primary hypothesis was that excessive inflammation would impact the adaptive immune response, a hypothesis we ended up rejecting and thus the novel findings here are primarily negative. We have evaluated well known predictors such as age, gender, cardiovascular markers in this cohort in other papers (PMID: 35691587). We have added serial testing in the immune response as figure 4. However, there was no perceivable difference in change in those that survived vs. those that died.

Added to methods:

“In a subset of patients, serial samples were obtained, totaling 395 specimens.”

“Differences in ACE2 inhibition between those that survived and died from serial testing was assessed using Wilcoxon rank sum test.”

Added to the results:

“Serial changes in ACE2 inhibition were assessed in 85 patients that survived and 20 patients that died (Figure 4). Patients that survived had a median increase in ACE2 inhibition of 1.63% per day (IQR: 0.22-4.7) and those that died had an increase in ACE2 inhibition of 2.32% per day (0.0-7.79, p = 0.5).”

Minor Comments:

1. Figure 1.

I prefer to present scatterplots to show the correlation between the factors, including "x," which displays columns. Also, Pearson's r value between 0.2 and 0.4 is generally considered a weak or poor correlation. Do not describe "significant correlation" only due to the p-value in correlation analysis, which is mostly misleading.

Thank you, we have modified significant to weak. We also appreciate the reviewer's feedback regarding the presentation of the data. The authors feel the current presentation is simpler and takes less space. However, we can present as three separate plots if the editor prefers.

2. The statistics method of Fig. 2/Sup. Fig. 2 needs to be included.

Thank you for pointing this out, the following was added to the methods:

“Pairwise comparisons were performed using Mann-Whitney U test.”

Re: Spectrum02459-24R1 (Systemic Inflammation is associated with worse outcomes from SARS-CoV-2 infection but not neutralizing antibody)

Dear Dr. Christopher W. Farnsworth:

Your manuscript has been accepted, and I am forwarding it to the ASM production staff for publication. Your paper will first be checked to make sure all elements meet the technical requirements. ASM staff will contact you if anything needs to be revised before copyediting and production can begin. Otherwise, you will be notified when your proofs are ready to be viewed.

Sincerely,
Jie Wang
Editor
Microbiology Spectrum